# The GEM-GECO Calcium Indicator Is Useable in *Ogataea parapolymorpha* Yeast, but Aggravates Effects of Increased Cytosolic Calcium Levels

**DOI:** 10.3390/ijms231710004

**Published:** 2022-09-02

**Authors:** Maria V. Kulakova, Azamat V. Karginov, Alexander I. Alexandrov, Michael O. Agaphonov

**Affiliations:** The Federal Research Center “Fundamentals of Biotechnology” of the Russian Academy of Sciences, Leninsky Prospect, 33, Build. 2, 119071 Moscow, Russia

**Keywords:** genetically encoded calcium indicator, yeast, calmodulin, Ca^2+^ ATPase, *Ogataea parapolymorpha*

## Abstract

Ca^2+^ is a ubiquitous second messenger, which allows eukaryotic cells to respond to external stimuli. The use of genetically encoded Ca^2+^ indicators allows real-time monitoring of cytosolic Ca^2+^ levels to study such responses. Here we explored the possibility of using the ratiometric Ca^2+^ indicator GEM-GECO for monitoring cytosolic Ca^2+^ concentration ([Ca^2+^]_cyt_) in the yeast *Ogataea parapolymorpha*. High-level production of GEM-GECO led to a severe growth defect in cells lacking the vacuolar Ca^2+^ ATPase Pmc1, which is involved in [Ca^2+^]_cyt_ control, and prompted a phenotype resembling that of Pmc1 deficiency, in a strain with wild-type *PMC1*. This was likely due to the presence of the calmodulin domain in GEM-GECO. In contrast to previous studies of genetically-encoded calcium indicators in neuronal cells, our results suggest that physiological effects of GEM-GECO expression in yeast cells are due not to Ca^2+^ depletion, but to excessive Ca^2+^ signaling. Despite these drawbacks, study of fluorescence in individual cells revealed switching of GEM-GECO from the Ca^2+^-free to Ca^2+^-bound state minutes after external addition of CaCl_2_. This was followed by gradual return of GEM-GECO to a Ca^2+^-free-state that was impaired in the *pmc1-Δ* mutant. These results demonstrate GEM-GECO usability for [Ca^2+^]_cyt_ monitoring in budding yeast.

## 1. Introduction

Ca^2+^ is a ubiquitous second messenger involved in the response to external stimuli in different types of living cells. In particular, it is involved in neuron activity, muscular contraction, cell cycle regulation, etc. [1,2,3,4]. Development of techniques for Ca^2+^ imaging in living cells has provided powerful tools to study its signaling roles.

Intracellular Ca^2+^ concentration can be monitored either by staining with fluorescent dyes (e.g., Fura or Fluo-2,3,4) or by using genetically encoded calcium sensors such as the Ca^2+^- coelenterazine-dependent luciferase aequorin [5,6] or artificially developed indicators based on fluorescent proteins (for a review see [7]). Among the latter, the most widely used type of genetically encoded Ca^2+^ indicators (GECI) is represented by circularly-permuted fluorescent proteins fused with calmodulin (CaM) at the C-terminus and possessing a Ca^2+^/CaM binding peptide at the N-terminus (reviewed in [7,8]). Efforts to improve their characteristics resulted in highly efficient GECIs (e.g., described in [9,10,11]) which are applicable for monitoring cytosolic Ca^2+^ concentration ([Ca^2+^]_cyt_) in different types of animal cells. One of these indicators was used to monitor [Ca^2+^]_cyt_ in dividing cells of the *Schizosaccharomyces pombe* yeast [4]. In such indicators, the Ca^2+^ dependent binding of the N-terminal peptide to the CaM domain leads to an alteration in the fluorescence parameters. As a rule, this is an increase in the fluorescence intensity; however, ratiometric GECI, which change either excitation (GEX-GECO) or emission (GEM-GECO) wavelength upon Ca^2+^ binding, have also been developed [9]. This property should allow monitoring [Ca^2+^]_cyt_ in individual cells independently of variation in the GECI production level. Notably, a GECI is effective in this kind of experiments if its presence in predominantly Ca^2+^-free form switches to predominantly Ca^2+^ -bound form within the range of the physiological [Ca^2+^]_cyt_ variations. This means that the K_d_ of its Ca^2+^-bound form must be between the resting [Ca^2+^]_cyt_ level and the level reached after stimulation.

Notably, high levels of GECI production were shown to affect neuronal cell physiology due to depletion of cytosolic Ca^2+^ [12], which may hamper its use in certain experiments. Although there is an example of the use of a GECI in fission yeast *Sch. pombe* [4], to our knowledge, no data are available on the effects of such proteins on yeast cell physiology and their applicability for [Ca^2+^]_cyt_ monitoring in various species of budding yeasts.

Similar to other eukaryotes, yeasts use Ca^2+^ as a second messenger allowing cells to respond to different external stimuli. This requires maintaining [Ca^2+^]_cyt_ at a low level (50–200 nM) by pumping it from the cytosol into intracellular depots. Studies of *Saccharomyces cerevisiae* have demonstrated that the main Ca^2+^ depot in the yeast cell is the vacuole, which accumulates Ca^2+^ due to action of the vacuolar calcium ATPase Pmc1 and the Ca^2+^/H^+^ exchanger Vcx1. The proton gradient, which empowers Vcx1 action, is generated by Vma1 (for a review see [13,14]).

Certain extracellular stimuli cause an increase in [Ca^2+^]_cyt_, activating the Ca^2+^ binding protein CaM, which interacts with CaM-dependent proteins, thus modulating their activity. One of these proteins is the protein phosphatase calcineurin which is activated both by binding Ca^2+^ directly and by interaction with Ca^2+^-bound CaM. Calcineurin dephosphorylates transcription factor Crz1 to trigger its translocation to the nucleus. This alters the expression of a number of genes including those involved in maintaining Ca^2+^ homeostasis [15], thus allowing the cell to restore [Ca^2+^]_cyt_ to the resting level.

Previously we have observed that inactivation of the vacuolar Ca^2+^ ATPase Pmc1 in methylotrophic yeast *Ogataea (Hansenula) polymorpha* and *O. parapolymorpha* has a wider impact on cell physiology than in the conventional yeast *S. cerevisiae*. In particular, besides the inability to grow at high concentration of Ca^2+^, in *Ogataea* yeasts this mutation causes hypersensitivity to sodium dodecyl sulfate (SDS), which is related to G_2_-M cell cycle transition arrest [16]. Here we used *O. parapolymorpha* to study whether the GEM-GECO indicator is suitable for [Ca^2+^]_cyt_ monitoring in this yeast and characterize the effects of its production on cell physiology in wild-type and *pmc1-Δ* strains.

## 2. Results and Discussion

### 2.1. GEM-GECO Expression Affects Cell Growth in a Pmc1-Dependent Manner

Efficient monitoring of Ca^2+^ using a genetically encoded fluorescent indicator such as GEM-GECO demands that its fluorescence be observable and quantifiable over the background fluorescence of the host cells. To achieve this in *O. parapolymorpha*, we used a codon-optimized gene coding for GEM-GECO, which was placed under control of a strong regulatable promoter of the *O. parapolymorpha MAL1* gene. The latter codes for maltase, which is required for utilization of some disaccharides including sucrose and is strongly induced upon growth on these carbon sources [17]. At the same time, utilization of sucrose supports growth of *O. parapolymorpha* as efficiently as that of glucose (our unpublished observation). In our study, this provides an advantage compared to the well-known promoters of methylotrophic yeasts, which are induced during utilization of the highly unfavorable carbon source methanol. The GEM-GECO expression cassette was placed into a vector possessing the G418 resistance marker and integrated into the *O. polymorpha* DL1-L genome. To select clones efficiently expressing GEM-GECO, the obtained transformants were grown on YP-Suc plates and irradiated by a 402 nm light emitting diode that excited GEM-GECO fluorescence. A clone with the brightest fluorescence which demonstrated high mitotic stability was designated DL5 and used in further experiments.

To test applicability of GEM-GECO for monitoring [Ca^2+^]_cyt_ in *O. parapolymorpha*, apart from the wild-type strain we also used a derivative of DL5 lacking the vacuolar Ca^2+^ ATPase Pmc1, which is responsible for maintaining [Ca^2+^]_cyt_ at a low level.

Presence of the GEM-GECO expression cassette did not noticeably affect cell growth in medium containing glucose, which represses the *MAL1* promoter. However, the culture of the wild-type strain possessing the GEM-GECO expression cassette did not reach the same density as the control strain lacking this cassette when cells were grown in sucrose-containing medium to stationary phase (Figure 1). Surprisingly, inactivation of *PMC1* noticeably decreased final cell density in sucrose-containing medium even in the absence of the GEM-GECO expression cassette, while its presence led to a deeper decrease in the final optical density of the *pmc1-Δ* mutant than in the strain with wild-type *PMC1* (Figure 1). Colonies of the *pmc1-Δ* mutant bearing the GEM-GECO expression cassette grew on sucrose-containing medium noticeably slower than those of the *pmc1-Δ* strain lacking this cassette (Figure 2A). This indicated that GEM-GECO expression aggravates some negative effects of the *pmc1-Δ* mutation. It was previously shown, that the Pmc1 deficiency in *O. polymorpha* and *O. parapolymorpha* causes hypersensitivity to SDS, which is related to the Hog1 and Wee1 dependent cell cycle regulation [16]. Although deletion of the *HOG1* gene itself affected cell growth, it reduced the negative effect of the GEM-GECO expression, since there was only small difference in stationary OD_600_ (Figure 1) and no noticeable difference in size of colonies (Figure 2A) between *hog1-Δ* mutant strains with and without the GEM-GECO expression cassette. The presence of the GEM-GECO expression cassette aggravated phenotypes of Pmc1 deficiency even when its promoter was repressed. Indeed, sensitivities to SDS and Ca^2+^ of the *pmc1-Δ* mutant possessing the GEM-GECO expression cassette were increased even on glucose-containing medium (Figure 2B). Expression of GEM-GECO in the strain with wild-type *PMC1* diminished the resistance to SDS (Figure 2C) resembling the manifestation of *PMC1* inactivation [16], which is most likely due to increased [Ca^2+^]_cyt_. This may indicate that GEM-GECO expression provokes activation of some Ca^2+^-dependent signaling pathway(s) that are usually triggered via increased, rather than depleted Ca^2+^ in the cytosol.

### 2.2. GEM-GECO Does Not Complement the Loss of Endogenous CaM in Ogataea Yeast

How could the presence of GEM-GECO cause induction of pathways usually triggered by increased [Ca^2+^]_cyt_? We proposed that the CaM domain of this protein somehow participates in Ca^2+^ dependent signaling. While it was previously shown that GECIs possessing this domain can affect cell physiology by depleting Ca^2+^ from cytosol [12], we suggested that this could be due to functioning of the GEM-GECO calmodulin domain in Ca^2+^-dependent regulation same as the endogenous calmodulin. The gene encoding CaM in *S. cerevisiae* (*CMD1*) is essential for viability [18] and we expected that it would also indispensable in *Ogataea* yeast. To explore whether GEM-GECO can complement the absence of the endogenous CaM in *O. parapolymorpha*, the DL5 strain (*leu2 P_MAL1_-GEM-GECO*) was transformed with a *CMD1* disruption cassette. PCR analysis of 24 transformants revealed integration of the disruption cassette into the *CMD1* locus in two of them; however, the wild-type locus was also detected in all cases (Appendix A). This was likely due to locus duplication or aneuploidy. Absence of transformants lacking the wild type *CMD1* gene strongly indicated that this gene is essential in *O. parapolymorpha,* as it is in *S. cerevisiae*. However, we were unable to test this via gene disruption in a *O. parapolymorpha* diploid strain since the DL-1 strain whose derivatives were used in this study, and commonly used in other laboratories, was an *O. parapolymorpha* asporogenous mutant [19]. To overcome this problem, we disrupted the *CMD1* gene in a diploid strain of *O. polymorpha*, which is the closest relative of this yeast. This diploid strain was obtained by crossing strains with and without the GEM-GECO expression cassette. The transformants with the *CMD1* disruption allele were incubated on sporulation medium and haploid segregants were selected using diethyl ether enrichment. Colonies of the segregants were obtained on both YPD and YP-Suc media, which provided repression or induction of GEM-GECO expression, respectively. None of the segregants possessed the disruption marker even though approximately half of them carried the expression cassette of GEM-GECO (Appendix A), which was expected to function as calmodulin. This proved that the *CMD1* gene is indispensable in *Ogataea* yeasts and that GEM-GECO does not rescue the absence of this gene. We hypothesize that the vital pathway depending on endogenous calmodulin and the pathway that exerts Ca^2+^-dependent SDS sensitivity are different, and that the latter pathway can react to both the endogenous calmodulin and GEM-GECO.

### 2.3. GEM-GECO Allows Monitoring Ca^2+^ Concentration in Cytosol in O. parapolymorpha

Although GEM-GECO expression affected cell growth in some cases, it did not cause cell death. This allowed us to study whether physiological changes in cytosolic Ca^2+^ concentration can be monitored by GEM-GECO in *O. parapolymorpha.* GEM-GECO fluorescence is excited at ~400 nm. The emission peak of the GEM-GECO Ca^2+^-free form is 513 nm, while its Ca^2+^-bound form emits with a peak at ~450 nm the quantum yield of the Ca^2+^-bound form at 450 nm is lower than that of Ca^2+^-free form at 513 nm [9]. Fluorescent microscopy analysis of cells expressing GEM-GECO revealed a cytosolic intracellular distribution pattern (Appendix A). We expected that inactivation of *PMC1* could lead to an increase in [Ca^2+^]_cyt_ that should increase fluorescence intensity at 450 nm and decrease it at 513 nm. Surprisingly we observed an increase in the fluorescence intensity in cell suspensions at both wavelengths in response to this mutation (Figure 3A). This could be due to higher GEM-GECO production level in the *pmc1-Δ* mutant.

To analyze GEM-GECO fluorescence in individual cells we used flowcytometry with a 405 nm excitation laser and fluorescence detection with 450 and 525 nm band-pass filters. Although the 450 nm fluorescence of majority of the GEM-GECO-expressing cells in regular conditions surpassed fluorescence of control cells lacking this expression cassette (Figure 3B), the autofluorescence at this wavelength was still considerable compared to the GEM-GECO signal. In contrast to 450 nm, the 525 nm fluorescence of GEM-GECO expressing cells strongly exceeded that of the control cells (Figure 3B). At the same time, when a considerable portion of GEM-GECO is in the Ca^2+^ bound form, which does not fluoresce at 525 nm, the auto fluorescence may noticeably distort the obtained results by overestimation of Ca^2+^-free form. To overcome this problem, calculated autofluorescence was subtracted from fluorescence intensities in individual cells (see Materials and Methods) and then the ratios between 450 nm and 525 nm fluorescence (FL_450_/FL_525_) were calculated as a characteristic of [Ca^2+^]_cyt_.

We suggested that a significant increase in external Ca^2+^ concentration would lead to some increase in [Ca^2+^]_cyt_. This would allow us to test applicability of GEM-GECO for monitoring [Ca^2+^]_cyt_ changes in yeast cells. To demonstrate this effect with an alternative method we used the fluorescent dye Fluo4-AM, which is often used to detect increases in intracellular [Ca^2+^] [20]. There was no detectable difference in Fluo-4 fluorescence between the stained and unstained cells incubated in the regular medium indicating that amount of Fluo-4 Ca^2+^-bound form is below the detection level. Five-minute incubation in the medium containing 100 mM CaCl_2_ led to noticeable increase in the cell fluorescence, which could be detected by fluorescent microscopy (Appendix A). At the same time incubation of the Fluo-4-loaded cells in CaCl_2_-supplemented medium led to their flocculation since a significant portion of events detected by flow cytometry had very high forward scattering (Appendix A). Nevertheless, analysis of events whose forward scattering corresponds to non-flocculating cells also showed a noticeable increase in the fluorescence (Appendix A).

Yeast cells defective in the vacuolar Ca^2+^ ATPase Pmc1 are unable to grow at high Ca^2+^ concentration in culture medium [16,21]. This indicates that increased Ca^2+^ concentration in culture medium should lead to some increase in [Ca^2+^]_cyt_ and that ability to resist it depends on Pmc1. Indeed, addition of 100 mM CaCl_2_ to the culture medium of strains expressing GEM-GECO led to a significant increase in FL_450_/FL_525_ in several minutes, which was followed by a decrease in this value during longer incubation in cells with wild-type *PMC1* gene (Figure 4A). In the *pmc1-Δ* mutant, similar increase in FL_450_/FL_525_ was also observed; however, this was not followed by a decrease in this median value. Moreover, a noticeable portion of cells continued to accumulate Ca^2+^ according to the increasing FL_450_/FL_525_ (Figure 4B). Yet, the peak value of FL_450_/FL_525_ in the population was somewhat reduced, which indicates that a portion of cells retain some ability to reduce [Ca^2+^]_cyt_ most likely due to action of the Vcx1 Ca^2+^/H^+^ exchanger. Notably, the FL_450_/FL_525_ value did not reach the original level after 1 h incubation even in the wild-type strain.

## 3. Materials and Methods

### 3.1. Yeast and Escherichia coli Strains

The *O. parapolymorpha* strains used in this study (Table 1) were constructed from the DL1-L strain (*leu2*) [22] or its derivative DLdaduA (*leu2 ade2-Δ ura3::ADE2*) [23]. The GEM-GECO expressing strain DL5 was obtained by transformation of DL1-L with the pKAM738 plasmid digested with BglII to facilitate its genome integration. The *PMC1* gene was inactivated in the *O. parapolymorpha* genome by replacement with either the *LEU2* gene as described in [16] or Cre recombinase recognition sequence *loxP* as described in [24]. EcoRI-digested empty vectors pCHLX [22] or pCCUR1 [23] possessing *LEU2* or *URA3* gene as a selectable marker, respectively, were integrated into a unidentified genome locus when complementing *leu2* or *ura3* auxotrophic mutations was required. The *O. polymorpha* diploid strain, in which the *CMD1* gene was inactivated, was obtained by crossing of 1B27-740G1M strain (*leu2 ade2 ura3::ADE2* {*P_MAL1_:GEM-GECO*}) possessing the GEM-GECO expression cassette integrated into unidentified locus with the 1B/SM/C2LX4 strain (*leu2 ade2 mox::uPA* [*LEU2*]) possessing an autonomously replicating plasmid with the *LEU2* selectable marker. The 1B27-740G1M strain was obtained by transformation of the 1B27 strain [16] with the EcoRV-cleaved pAM740 GEM-GECO expression vector, whose *LEU2*-containing backbone was then excised by Cre-recombination. The 1B/SM strain was obtained from the 1B strain [25] by replacement of the *MOX* gene with the uPA expression cassette as described previously [26]. Then the obtained strain was transformed with an autonomously replicating plasmid pC2LX4. The diploid strain was selected on leucine, uracil and adenine omission medium. To obtain a *leu2* auxotroph, which could be transformed with the *CMD1* disruption cassette, the autonomously-replicating plasmid was lost from the obtained diploid. The *E. coli* DH-5α strain was used in plasmid construction procedures.

### 3.2. Culture Media and Transformation Procedures

Complex media contained 2% Peptone, 1% Yeast extract and 2% glucose (YPD) or 1% sucrose (YP-Suc) as a carbon source. The synthetic medium SC-D (2% glucose, 0.67% Yeast Nitrogen Base with ammonium sulfate) was used for selection of transformants obtained with an auxotrophic selectable marker. The LB medium (1% Tryptone, 0.5% Yeast extract, 1% NaCl) was used to cultivate *E. coli*. *E. coli* was transformed with plasmids by method described in [27]. Yeast cells were transformed by Li-acetate method [28] with some modifications [29] as follows. Cells from 300 μL of an exponentially grown culture were harvested by centrifugation in a bench-top microcentrifuge (Eppendorf, Hamburg, Germany) at 5000 rpm for 30 sec, washed and re-suspended in 42 μL of sterile water. 2 μL of the DNA-carrier solution (10 mg/mL, shared and denatured by boiling) and 6 drops (approx. 90 μL) of 70% PEG 4000 were added and mixed well. After that the suspension was mixed with 9 ul of 1 M Li-acetate solution and dispensed by 20–23 μL to add 1 μL of transforming DNA to each portion. Suspensions were incubated first at 30 °C for 30 min than at 45 °C for 30 min. Cells were washed with YPD medium and spread on plates for selection of transformants.

### 3.3. Plasmid Construction

The DNA fragment with codon-optimized sequence encoding GEM-GECO was synthesized by Biomatik Corporation (Kitchener, ON, Canada). This fragment was fused with *O. parapolymorpha MAL1* promoter sequence (EcoRV-SspI 579 bp fragment of PCR product obtained with primers AGGAATTAATATTGTCAAGAGGG and TGGACACGCGTGTGTCGAGAA) and inserted between PvuII and CfrI sites of the pKAM555 vector [30] to obtain pKAM738 plasmid. This vector possesses a selectable marker, which allows selection of transformants by kanamycin resistance in *E. coli* and G418 resistance in yeast. The pAM740 GEM-GECO expression vector was obtained by insertion of a 1.6 kb EcoRI-NheI fragment of pKAM738 between EcoRI-SpeI sites of the pAM631 self-excisable vector [24].

### 3.4. Analysis of GEM-GECO and Fluo-4 Fluorescence

GEM-GECO production in yeast transformants was induced by growing cells in the YP-Suc medium. Flow cytometric analysis of fluorescence (FL) in individual cells was performed using the CytoFLEX S flow cytometer (Beckman Coulter, Pasadena, CA, USA). The GEM-GECO fluorescence was excited by a 405 nm laser and measured using 450/20 nm (FL_450_) and 525/40 nm (FL_525_) bandpass filters. 10,000 cells were acquired in each sample. Cell fluorescence excited by 488 nm laser and measured using 525/40 nm bandpass filters (“FITC channel”) was found not to depend on presence of GEM-GECO. At the same time there was a good correlation between the “FITC channel” fluorescence and 405 nm-excited 525 nm and 450 nm autofluorescence (Appendix A). This allowed us to use the “FITC channel” fluorescence to estimate autofluorescence in GEM-GECO-expressing cells. To do this, trend lines of dependence of autofluorescences at 525 nm and 450 nm on “FITC channel” fluorescence in strains lacking GEM-GECO were calculated using MS Excel. Based on the obtained functions and “FITC channel” fluorescence values in the strains producing GEM-GECO, values of autofluorescence were calculated for individual cells and subtracted from the measured 525 nm and 450 nm fluorescence values. The obtained values were normalized to autofluorescence at each wavelength. The FL_450_/FL_525_ ratio was used as a characteristic of [Ca^2+^]_cyt_ [9]. Fluorescence of cell suspensions was measured using an EnSpire plate reader (Perkin Elmer, Waltham, MA, USA). The obtained values were normalized to the total cell protein, which was assayed using the biuretic method described elsewhere [31].

Loading cells with Fluo-4 was performed according to the published protocol [32] with minor modifications as follows. Yeast cells were grown in YPD to OD_600_ between 1.0 and 2.0, washed twice with PBS (137 mM NaCl, 2.7 mM KCl, 8 mM Na_2_HPO_4_, and 2 mM KH_2_PO_4_) and resuspended in PBS to achieve the original cell density. The cell suspension was supplemented with 3 μM Fluo-4-AM (Thermo Fisher Scientific, Waltham, MA, USA) and incubated for 1 h at 37 °C. To test effect of external Ca^2+^ concentration rise, the cell suspension was diluted either with regular YPD, or YPD supplemented with 100 mM CaCl_2_. In the case of microscopic imaging, the cell suspension was loaded onto solid medium pads [33] made up of regular YPD, or YPD supplemented with 100 mM CaCl_2_.

## 4. Conclusions

In this work we have studied the applicability of the ratiometric GEM-GECO Ca^2+^ indicator for monitoring [Ca^2+^]_cyt_ in yeast *O. parapolymorpha*. Production of this protein in cells with wild type Ca^2+^ homeostasis had a negligible effect on cell physiology. However, inactivation of the vacuolar Ca^2+^ ion pump Pmc1 manifested a pronounced interaction with GEM-GECO expression, which was revealed as aggravation of *pmc1-Δ* mutant phenotypes. Although previous studies indicated that physiological effects of such indicators in neuronal cells are related to Ca^2+^ depletion in cytosol, the effects we observed here were more likely to be related to an enhanced response to [Ca^2+^]_cyt_ rise. We hypothesize that this is due to the CaM domain of GEM-GECO, which might interact with the host Ca^2+^-dependent machinery. Nevertheless, this indicator is applicable for monitoring [Ca^2+^]_cyt_ in yeast cells. In particular, it is predominantly present in the Ca^2+^-free form in cells during normal conditions and converts to the Ca^2+^-bound form when [Ca^2+^]_cyt_ rises in response to external Ca^2+^ concentration rise. This means that the [Ca^2+^]_cyt_ in yeast cells is essentially below the K_d_ of the GEM-GECO-Ca^2+^ complex and can rise above this value in response to various treatments. We also observed that GEM-GECO expression under control of *MAL1* promoter somehow depended on Ca^2+^ homeostasis, since it became elevated after inactivation of the *PMC1* gene encoding the vacuolar Ca^2+^ ATPase. However, this did not affect [Ca^2+^]_cyt_ monitoring since this indicator is ratiometric.

## Figures and Tables

**Figure 1 ijms-23-10004-f001:**
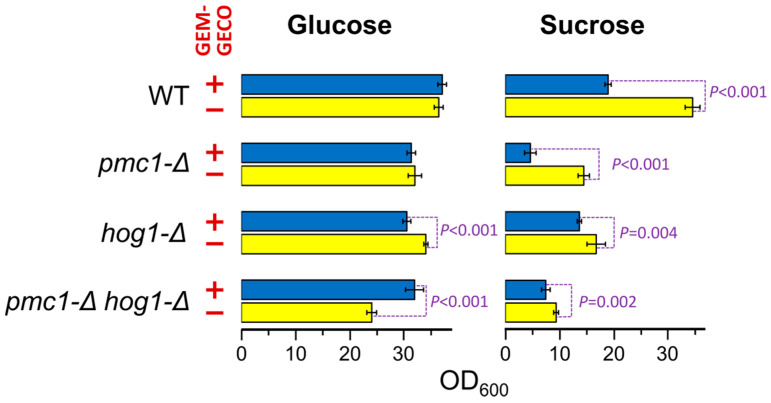
Effect of GEM-GECO expression on OD_600_ of saturated cultures of strains bearing wild-type or deletion alleles of the *PMC1* and *HOG1* genes. Overnight YPD cultures were 100-fold diluted with YPD (Glucose) or YP-Suc (Sucrose) medium and incubated at 37 °C for 24 h. WT, DL5-LC (+) and DL1-LC (−) strains; *pmc1-Δ*, DL5-pmc1-LC (+) and DL-pmc1-LC (−) strains; *hog1-Δ*, DL5-hog1 (+) and DLdaduA-hog1-UC (−) strains; *pmc1-Δ hog1-Δ,* DL5-pmc1-hog1 (+) and DLdaduA-pmc1-hog1 (−) strains. The means with standard deviations and *P*-values were calculated from data obtained in six replicates.

**Figure 2 ijms-23-10004-f002:**
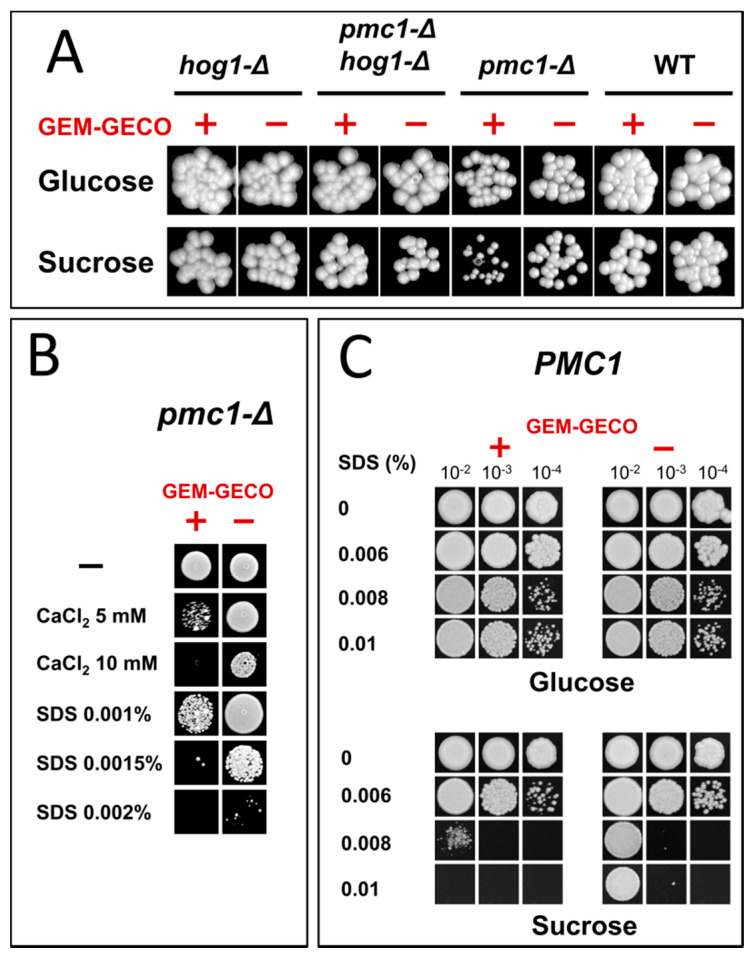
Growth of strains differing in the expression of the GEM-GECO cassette on YPD (Glucose) and YP-Suc (Sucrose) solid media. (**A**), Spotting of 10^−4^-fold diluted YPD cultures on regular YPD and YP-Suc. WT, DL1-LC and DL5-LC strains; *pmc1-Δ*, DL5-pmc1-LC and DLdaduA-pmc1-LC strains; *hog1-Δ*, DL5-hog1 and DLdaduA-hog1-UC strains; *pmc1-Δ hog1-Δ*, DL5-pmc1-hog1 and DLdaduA-pmc1-hog1 strains. (**B**), Spotting of 10^−4^-fold diluted YPD cultures of DL-pmc1 (GEM-GECO −) and DL5-pmc1 (GEM-GECO +) on YPD with indicated supplements. (**C**), Spotting of 10^−2^-, 10^−3^-, and 10^−4^-fold diluted YPD cultures of DL1-L (GEM-GECO −) and DL5 (GEM-GECO +) on YPD and YP-Suc with indicated supplements.

**Figure 3 ijms-23-10004-f003:**
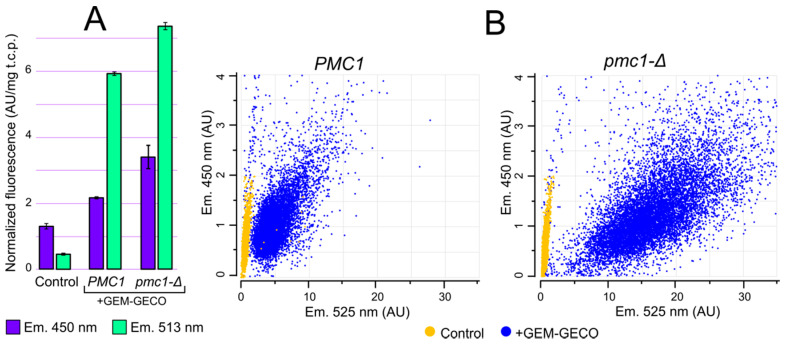
Comparison of fluorescence characteristics of cells with (+GEM-GECO) and without (Control) the GEM-GECO expression cassette. (**A**), Fluorescence normalized to the total cell protein in the cell suspensions of DL1-LC (Control), DL5-LC (*PMC1*) and DL5-pmc1-LC (*pmc1-Δ*) strains. The fluorescence was excited by 395 nm light. (**B**), Flowcytometric analysis of cell fluorescence of strains: DL1-LC (Control, *PMC1*), DL1-pmc1-LC (Control, *pmc1-Δ*), DL5-LC (*PMC1*) and DL5-pmc1-LC (*pmc1-Δ*) strains (450 nm and 525 nm emission, 405 nm excitation).

**Figure 4 ijms-23-10004-f004:**
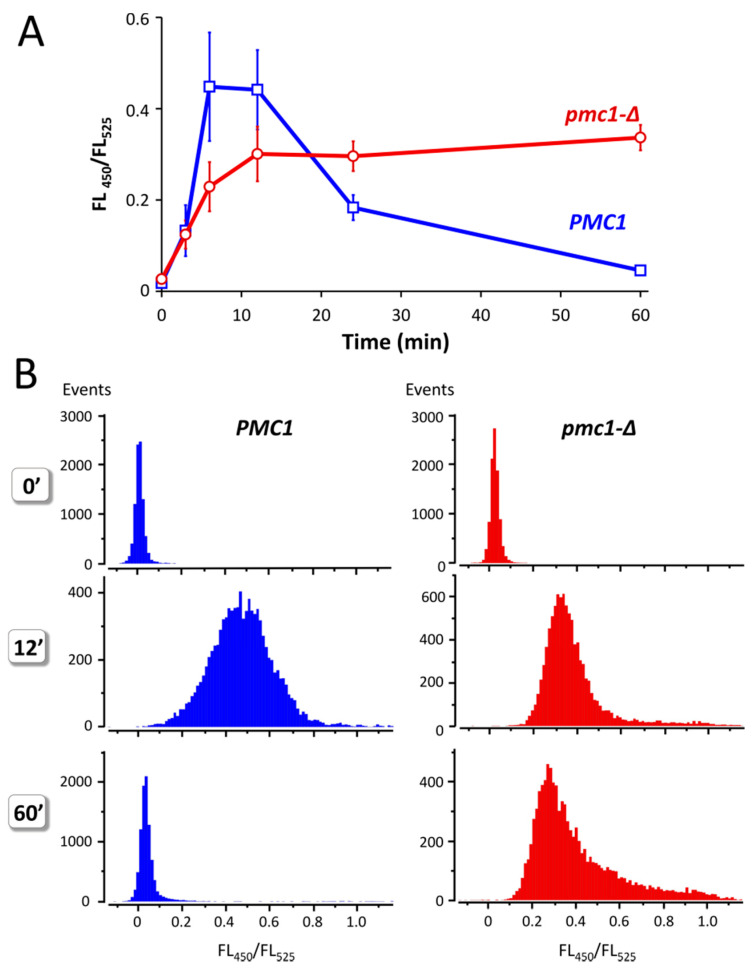
Effect of Ca^2+^ concentration rise in culture medium on the ratio between fluorescence at 450 nm and 525 nm (FL_450_/FL_525_) excited by 405 nm laser in individual cells of strains DL5 (*PMC1*) and DL5-pmc1 (*pmc1-Δ*) producing GEM-GECO. Overnight YP-Suc cultures were 3-fold diluted with fresh YP-Suc medium, grown for 6 h. The obtained cultures were diluted 120-fold with YP-Suc medium supplemented with 100 mM CaCl_2_. Fluorescence in individual cells was measured using flowcytometry after 3′, 6′, 12′ and 60′ incubation. The 0′ samples are represented by the same cultures, which were diluted by YP-Suc without the CaCl_2_ supplement. (**A**), Dynamics of FL_450_/FL_525_ median values during incubation with 100 mM CaCl_2_ in the culture medium. Mean values of medians with standard deviations of these means were calculated using data from five and six independent replicates in the *pmc1-Δ* mutant and wild-type strain, respectively. (**B**), distribution of cells with noted FL_450_/FL_525_ values in cultures before (0′) and after 12′ and 60′ incubation in the culture medium supplemented with 100 mM CaCl_2_.

**Table 1 ijms-23-10004-t001:** *O. parapolymorpha* strains used in this study. Names of plasmid genes integrated into unidentified genome locus as a part of a plasmid are shown in brackets.

Strain	Genotype
DL1-L	*leu2*
DLdaduA	*leu2 ade2-Δ* *ura3::ADE2*
DL1-LC	*leu2 {LEU2}*
DL-pmc1-LC	*leu2 pmc1::loxP {LEU2}*
DL5	*leu2 {P_MAL1_-GEM-GECO}*
DL5-LC	*leu2 {P_MAL1_-GEM-GECO} {LEU2}*
DL5-pmc1-LC	*leu2 pmc1::loxP {P_MAL1_-GEM-GECO} {LEU2}*
DL5-pmc1-hog1	*leu2 pmc1::loxP hog1::LEU2 {P_MAL1_-GEM-GECO}*
DL5-hog1	*leu2 hog1::LEU2{P_MAL1_-GEM-GECO}*
DLdaduAU-LC	*leu2 ade2-Δ* *ura3::ADE2 {LEU2} {URA3}*
DLdaduA-pmc1-LC	*leu2 ade2-Δ* *ura3::ADE2 pmc1::URA3 {LEU2}*
DLdaduA-hog1-UC	*leu2 ade2-Δ* *ura3::ADE2 hog1::LEU2 {URA3}*
DLdaduA-pmc1-hog1	*leu2 ade2-Δ* *ura3::ADE2 pmc1::URA3 hog1::LEU2*

## Data Availability

All data generated or analyzed during this study are included in this published article.

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
