# Peer review of "The GEM-GECO Calcium Indicator Is Useable in Ogataea parapolymorpha Yeast, but Aggravates Effects of Increased Cytosolic Calcium Levels"

_ijms, 2022, doi:10.3390/ijms231710004_

Round 1

Reviewer 1 Report

The manuscript entitled " The GEM-GECO calcium indicator is useable in Ogataea para-polymorpha yeast, but aggravates effects of increased cytosolic calcium levels" has tried to demonstrate the usability of GEM-GECO as a calcium sensor in yeast. The authors have shown that the expression of GEM-GECO can have diverse effects on yeast. However, this manuscript lacks proper direction to be scientifically sound. The authors tried to develop a story with a small amount of preliminary dataset, but this work is still in the naïve stage of being published.

1.             The abstract, introductions and results flow lack a proper understanding of the major scientific questions the authors tries to address. It's not clear from their study what are the

a.       primary goals of this study

b.       unsolved questions or what are the main questions they want to answer

c.       the conclusions are unclear, and the authors have overinterpreted their data without using enough control samples.

d.       The authors do not provide enough mechanistic insights about their different results.

2.             As the author's main focus of the study is GEM-GECO sensor protein, they need to provide a detailed schematic of this system regarding how they work in the cellular system and are used for calcium sensing.

3.             The authors have mentioned genetically encoded fluorescent Ca2+ indicators as GECI. However, it should be genetically encoded Ca2+ indicators (GECI). This may confuse the readers. Although there are many kinds of GECIs reported in the literature, the authors didn't mention clearly in the introduction which fluorescent sensor they used in their study and what are the specific characteristics of that GECI.

4.             The last part of the introduction is very scattered. Line 63-69 have very little link to their further study. As I mentioned above, the intro does not boil down to a specific set of goals or questions and does not clarify the motivation behind this study. The authors should clarify this. Why the authors choose O. parapolymorpha over Ogataea is not clear.

5.             Why figure 1 can at all be important here? There is no error bar in the data. How do the glucose and sucrose media cell growth experiments contribute to the general understanding or goals of this study? It seems out of context and does not lead to any better mechanistic understanding of the system.

6.             In the Glucose media, the expression of GEM-GECO does not show any significant changes. Only major effect comes when the glucose media vs. sucrose media is compared. So under the glucose media condition, the sensor can be studied, and glucose vs. sucrose media can have more complicated pathways to follow.

7.             One of the main conclusions the authors like to draw from their study is that the expression of GEM-GECO leads to an increase in cytosolic calcium concentration rather than a decrease. However, the authors have not provided any direct evidence. The conclusion from the fig1 and fig 2 is very much superficial, and there could be many factors that may lead to the differential growth effect the authors have observed. For example, one of the possibilities is the different expression levels of the proteins. The authors have not quantified the expression levels of the different proteins in different mutated systems in the O. parapolymorpha. It could be possible that GEM-GECO, hog1, pmc1 all have different expression levels in these mutants used in fig 1 and fig 2, which leads to differential effects.

8.             The authors should show data to confirm the deletion of the pmc1, hog1 genes by western blotting or other complementary techniques. 

9.             In section 2.2, the authors tried to claim that GEM-GECO mimics the CaM protein. The authors should perform an experiment where they over-express CaM to determine if the outcomes are similar to the expression of GEM-GECO.

10.         There are many cases the authors narrated different results but didn't show any data. Some of the examples are,

a.       PCR analysis revealed the disruption allele in some transformants, however the wild-type locus was also detected in all cases.

b.       Colonies of the segregants were obtained on both YPD and YP-Suc media, which provided repression or induction of GEM-GECO expression, respectively. None of the segregants possessed the disruption marker even though approximately half of them carried the expression cassette of GEM-GECO, which was expected to function as calmodulin.

11.         The authors should use another non-yeast cell line to show that the effect of pmc1 deletion with GEM-GECO is general or very specific to the cell line they have used.

12.         The authors didn't write enough explanations explaining the patterns of fig 4a.

13.         Authors should provide confocal images of the yeast cell expressing GEM-GECO to show how the fluoresce from the cells change at different Calcium concentrations. The images should be plotted in the same look-up table for visual quantification.

14.         It has been shown that SERCA1a expression restored the growth of pmc1 mutants in media containing high Ca2+ concentrations, Degand et al., Molecular Microbiology (1999)31(2), 545–556. The authors should use this strategy as a control.

15.         The main conclusions in this paper are still speculative. The authors need to use complimentary techniques to support the conclusions. 

Author Response

Q: 1.             The abstract, introductions and results flow lack a proper understanding of the major scientific questions the authors tries to address. It's not clear from their study what are the

  1. primary goals of this study
  2. unsolved questions or what are the main questions they want to answer
  3. the conclusions are unclear, and the authors have overinterpreted their data without using enough control samples.
  4. The authors do not provide enough mechanistic insights about their different results.

A: We strongly disagree with these statements.

Q: 2.             As the author's main focus of the study is GEM-GECO sensor protein, they need to provide a detailed schematic of this system regarding how they work in the cellular system and are used for calcium sensing.

A: This sensor has been developed previously and its properties have been already described. This study with corresponding reference is mentioned in the manuscript. Mechanistic principals of GECIs action are briefly described in the introduction with corresponding references.

Q: 3.             The authors have mentioned genetically encoded fluorescent Ca2+ indicators as GECI. However, it should be genetically encoded Ca2+ indicators (GECI). This may confuse the readers. Although there are many kinds of GECIs reported in the literature, the authors didn't mention clearly in the introduction which fluorescent sensor they used in their study and what are the specific characteristics of that GECI.

A: This sentence was rephrased.

Q: 4.             The last part of the introduction is very scattered. Line 63-69 have very little link to their further study. As I mentioned above, the intro does not boil down to a specific set of goals or questions and does not clarify the motivation behind this study. The authors should clarify this.

A: We have corrected some errors in abbreviation in this section, however overall, it was required for providing the reader with a brief overview of the Ca-dependent signaling pathways in yeast, some components of which (CaM) will be mentioned in the results section. We feel that the final phrase of the introduction sums up the main goal of the study “Here we used O. parapolymorpha to study whether the GEM-GECO indicator is suitable for [Ca2+]cyt monitoring in this yeast and characterize the effects of its production on cell physiology in wild-type and pmc1-Δ strains.”

Q: Why the authors choose O. parapolymorpha over Ogataea is not clear.

A: O. polymorpha and O. parapolymorpha are very closely related species and there were no crucial advantages in choosing one or the other. One important parameter is that O. parapolymorpha is more accessible for targeted gene inactivation than O. polymorpha due to higher frequency of integration of transforming DNA via homologous recombination. Beside this, in preliminary experiments, we could achieve somewhat higher GEM-GECO production in O. parapolymorpha transformants, which helped us at the initial steps of the study. However these facts seem to us unimportant to be mentioned in the manuscript.

Q: 5.             Why figure 1 can at all be important here? There is no error bar in the data. How do the glucose and sucrose media cell growth experiments contribute to the general understanding or goals of this study? It seems out of context and does not lead to any better mechanistic understanding of the system.

A: The reviewer is mistaken. Error bars are present in the figure. The use of glucose and sucrose as a carbon source is crucially important since the GEM-GECO expression cassette is under control of the MAL1 promoter, which is strongly induced during growth on sucrose and repressed on glucose. This is explained in the manuscript. Thus, figure 1 is central for showing the growth defects caused by sensor production.

Q: 6.             In the Glucose media, the expression of GEM-GECO does not show any significant changes. Only major effect comes when the glucose media vs. sucrose media is compared. So under the glucose media condition, the sensor can be studied, and glucose vs. sucrose media can have more complicated pathways to follow.

A: Glucose represses the promoter of the GEM-GECO expression cassette, which is why there are no effects on growth rate. Growth in medium with glucose was used as a negative control, while medium with sucrose revealed the effects of GEM-GECO.  

Q: 7.             One of the main conclusions the authors like to draw from their study is that the expression of GEM-GECO leads to an increase in cytosolic calcium concentration rather than a decrease. However, the authors have not provided any direct evidence. The conclusion from the fig1 and fig 2 is very much superficial, and there could be many factors that may lead to the differential growth effect the authors have observed. For example, one of the possibilities is the different expression levels of the proteins. The authors have not quantified the expression levels of the different proteins in different mutated systems in the O. parapolymorpha. It could be possible that GEM-GECO, hog1, pmc1 all have different expression levels in these mutants used in fig 1 and fig 2, which leads to differential effects.

A: We are afraid that the reviewer misunderstood our statements. We did not claim that GEM-GECO increases cytosolic calcium concentration. We assert that effects of GEM-GECO production are unlikely to be due to calcium chelation by the sensor, but may result from INCREASED SENSITIVITY to calcium rise in the cytosol. The most likely explanation is that the GEM-GECO calmodulin domain is partially functional in signal transduction in yeast. This is discussed in the manuscript.    

The level of most of the noted proteins is irrelevant, because we only compare deletion strains to wild type strains. The reviewer is somewhat right in mentioning the level of GEM-GECO as a possibly important factor, because its level is somewhat elevated in the pmc1 mutant (see figure 3). However, phenotypic effects of GEM-GECO  were observed both in WT and pmc1 strains.

Q: 8.             The authors should show data to confirm the deletion of the pmc1, hog1 genes by western blotting or other complementary techniques.

A: Clones with deletions of these genes were selected by phenotypic characteristics and confirmed by PCR analysis as we did in our previously published studies. This is a routine procedure and we do not think that presenting pictures of agarose gels with bands of PCR fragments can help understand our manuscript.

Q: 9.             In section 2.2, the authors tried to claim that GEM-GECO mimics the CaM protein. The authors should perform an experiment where they over-express CaM to determine if the outcomes are similar to the expression of GEM-GECO.

A: Actually GEM-GECO contains a CaM domain. We do not claim but suggest that it can be partially functional in yeast. Demonstration that overexpression of CaM can lead to effects that are similar to those of GEM-GECO expression was not the goal of our study, also the results would be difficult to interpret due to the multiple number of pathways affected by CaM.

  1. There are many cases the authors narrated different results but didn't show any data. Some of the examples are,

  1. PCR analysis revealed the disruption allele in some transformants, however the wild-type locus was also detected in all cases.

  1. Colonies of the segregants were obtained on both YPD and YP-Suc media, which provided repression or induction of GEM-GECO expression, respectively. None of the segregants possessed the disruption marker even though approximately half of them carried the expression cassette of GEM-GECO, which was expected to function as calmodulin.

A: Photographs of gels with PCR analysis of gene disruptions in yeast are usually not presented in research papers. Nevertheless, to follow the reviewer’s request we included these data in supplementary materials together with data on analysis of phenotype segregation in haploid segregants of the CMD1-disrupted diploid (Tables S1 and S2).

Q: 11.         The authors should use another non-yeast cell line to show that the effect of pmc1 deletion with GEM-GECO is general or very specific to the cell line they have used.

A: Our goal was to study effects and applicability of GEM-GECO in the Ogataea yeast. Particular manifestations of misbalanced calcium-dependent signaling can be very different in other organisms. But our results can be taken into account by researchers studying cytosolic calcium dynamics in cells of other organisms.

Q: 12.         The authors didn't write enough explanations explaining the patterns of fig 4a.

A: The figure legend was edited, also taking into account the request of Reviewer 2.

  1. Authors should provide confocal images of the yeast cell expressing GEM-GECO to show how the fluoresce from the cells change at different Calcium concentrations. The images should be plotted in the same look-up table for visual quantification.

A: The fluorescence of Ca2+ -free GEM-GECO is much brighter than that of its Ca2+ -bound form. Fluorescent microscopy analysis of the GEM-GECO switch between Ca2+ -bound and Ca2+ - free state requires a very specific set of filters, which can efficiently remove fluorescence of the Ca2+ -free form during visualization of the Ca2+ -bound form. Unfortunately, we do not have the proper filter set on our microscope and cannot obtain it rapidly. Nevertheless, we have performed microscopic imaging of GEM-GECO fluorescence, which indicated its cytosolic localization. The microphotographs are included into the supplementary materials.

Q: 14.         It has been shown that SERCA1a expression restored the growth of pmc1 mutants in media containing high Ca2+ concentrations, Degand et al., Molecular Microbiology (1999)31(2), 545–556. The authors should use this strategy as a control.

A: Yeasts do not have an ortholog of the SERCA pump. We do not understand how expression of this heterologous protein can serve as a control in our experiments. The strains with wild-type PMC1 serve as controls for pmc1 mutant strains.

Q: 15.         The main conclusions in this paper are still speculative. The authors need to use complimentary techniques to support the conclusions.

A: We disagree with this statement.

Reviewer 2 Report

In the manuscript, authors applied ratiometric genetically-encoded calcium indicator (GEM-GECO) in methylotrophic yeast. They carefully investigated the effect of the expression of sensor on yeast cell physiology, indicating the biosensor somehow affects cellular growth and Ca2+ tolerance.  Yet, this sensor seems to enable monitoring Ca2+ dynamics in the yeast with a good calibration of autofluorescence. Overall, their goal to apply the Ca2+ indicator in the yeast has been essentially achieved. Though this manuscript is worth publishing, some minor concerns listed below should be cleared before publication.

1. They reasoned that the growth defect by GEM-GECO comes from the activation of Ca2+ signaling pathways, lines 122, 173 ,244. However the data just showed the similarities of growth defect to pmc1 deletion. Direct observation of the Ca2+ signaling activation in GECO expressing cell is required to argue this statement, or they should replace these sentence with weaker words.

2. In the legend of Figure 2C, they should add YP-Suc.

3. It is not clear which statistic values are represented in Fig 4A. They mentioned that they calculated medial values using independent replicates and showed the medial values with standard deviation. Are the medial values the median of each cell, or each replicate? Moreover, they showed medial values with standard deviation, but standard deviation can only be calculated with mean values because its definition is the summary of the relative distance from mean values. That's why plots in Fig4A should represent the mean values (of cells or replicates?), or different statistics (generally quartile) should be used if they are indeed the medial values.

Author Response

Q: 1. They reasoned that the growth defect by GEM-GECO comes from the activation of Ca2+ signaling pathways, lines 122, 173 ,244. However the data just showed the similarities of growth defect to pmc1 deletion. Direct observation of the Ca2+ signaling activation in GECO expressing cell is required to argue this statement, or they should replace these sentence with weaker words.

A: We have modified the wording, using milder expressions.

Q: 2. In the legend of Figure 2C, they should add YP-Suc.

A: This sentence was corrected. We thank the reviewer for noting this error.

Q: 3. It is not clear which statistic values are represented in Fig 4A. They mentioned that they calculated medial values using independent replicates and showed the medial values with standard deviation. Are the medial values the median of each cell, or each replicate? Moreover, they showed medial values with standard deviation, but standard deviation can only be calculated with mean values because its definition is the summary of the relative distance from mean values. That's why plots in Fig4A should represent the mean values (of cells or replicates?), or different statistics (generally quartile) should be used if they are indeed the medial values.

A: The presented values were mean (average) values of FL450/FL525 medians obtained in different repeats of the experiment. This sentence was re-written to be unambiguous.

Round 2

Reviewer 1 Report

The authors have tried to address many of the concerns of their work; however, the authors didn’t attempt to clarify the main concerns regarding their work. In the present version, the work seems to be incomplete and lacks novelty. The main concerns remain similar; as I stated earlier,

  1. The main goal as far as the authors tried to state whether GEM-GECO is a suitable biosensor for [Ca2+]cyt in yeast. However, the authors have not quantified the Ca2+ concentration in the cytoplasm by any alternative methods. This is required to validate their results.

“our results show that physiological effects of GEM-GECO expression in yeast cells are due not to Ca2+ depletion, but to excessive Ca2+ signaling.” One of the main conclusions of their work comes from indirect ways. The authors need to show by using alternative quantitative methods that actually the cytoplasmic Ca2+ concentration increases or decreases in the cytoplasm.

  1. The authors do not provide enough mechanistic insights about their different results. For example, how GEM-GECO production and partial function of the CaM domain in GEM-GECO leads to INCREASE SENSITIVITY to calcium rise in the cytosol is unclear. The authors should perform further experiments to support their statements.
  2. The authors should not state the hypothesis in the conclusion. If the authors are stating a hypothesis for their work, they should better test that.

In question 5, one of my questions was wrong. I apologize for this. By mistake, I mentioned that there was no error bar. The error bar was present, but what I actually wanted to mention, which I missed, the authors have not shown any statistical significance of their data interpretations. The authors need to provide p values for Fig 1.

Author Response

Q: 1.      The main goal as far as the authors tried to state whether GEM-GECO is a suitable biosensor for [Ca2+]cyt in yeast. However, the authors have not quantified the Ca2+ concentration in the cytoplasm by any alternative methods. This is required to validate their results.

“our results show that physiological effects of GEM-GECO expression in yeast cells are due not to Ca2+ depletion, but to excessive Ca2+ signaling.” One of the main conclusions of their work comes from indirect ways. The authors need to show by using alternative quantitative methods that actually the cytoplasmic Ca2+ concentration increases or decreases in the cytoplasm.

A: Except the use of GECI, there are two another approaches to monitoring cytosolic Ca2+ described so far and mentioned in Introduction. Namely, staining cells with special fluorescent dyes and the use of Ca2+- and coelenterazine-dependent luciferase aequorin. The former approach requires loading cells with the dye while the latter one requires loading cells with coelenterazine, which is not synthesized by yeast cells. Moreover, the aequorin-based approach cannot be used for individual cells analysis. Precise quantification of cytosolic Ca2+ by these methods requires the use of Ca2+ ionophores. We have preliminary data, that the conventional ionophores A23187 and ionomycin are not sufficiently efficient to rapidly equalize internal and external Ca2+ concentration at least in Ogataea cells. We are going to publish these results soon. At the same time the GEM-GECO-based approach does not require this step, since GEM-GECO switches between predominantly Ca2+ -free and Ca2+ -bound form when Ca2+ concentration traverses through the Kd value, which has been previously determined (reference 9). The ratio of fluorescence intensity of these forms is proportional to the Ca2+ concentration in a range close to the Kd value and does not depend on the GEM-GECO expression level. Most often, precise quantification of Ca2+ concentration is not required in experiments studying Ca2+ intracellular signaling. It is more important whether change of concentration falls into the dynamic range of the detection system.  This is what we actually observed in our experiments. Within its dynamic range, GEM-GECO theoretically should give much more precise estimation of the [Ca2+ ]cyt than the other noted methods using fluorescent dyes or aequorin. We have actually tried using these approaches and found them much less convenient and interpretable for our purposes than the use of GEM-GECO. Nevertheless, to follow the reviewer’s request we have confirmed the effect of external Ca2+ rise on its intracellularconcentration using the Fluo-4-AM fluorescent dye. These data are discussed in the manuscript and presented as supplementary material (Figures S3 and S4). 

Regarding the mentioned phrase from the abstract, we do not state that this is one of the main conclusions of our study. We agree that this was not proved definitively. That was why we discuss this in milder expressions in the manuscript. This sentence was corrected to express this idea in a less assertive phrasing.

  1. The authors do not provide enough mechanistic insights about their different results. For example, how GEM-GECO production and partial function of the CaM domain in GEM-GECO leads to INCREASE SENSITIVITY to calcium rise in the cytosol is unclear. The authors should perform further experiments to support their statements.

A: This was a hypothesis drawn from the obtained results. Proving it was not the goal of our study. Actually, we do not even have the means to test it at present.

  1. The authors should not state the hypothesis in the conclusion. If the authors are stating a hypothesis for their work, they should better test that.

A: This hypothesis was based on the obtained result and explains them. It points to a possible mechanism of GEM-GECO action in the cell, which can be important for researchers that are going to use this indicator. Proving this hypothesis was not the goal of this study. We disagree with the reviewer that a hypothesis explaining our results cannot be mentioned in the Conclusion section, if it is clearly labelled a hypothesis. If it would be tested and proved, it would no longer be a hypothesis, but a fact.

Q: In question 5, one of my questions was wrong. I apologize for this. By mistake, I mentioned that there was no error bar. The error bar was present, but what I actually wanted to mention, which I missed, the authors have not shown any statistical significance of their data interpretations. The authors need to provide p values for Fig 1.

A: We repeated this experiment with more replicates and presented p-values in the Figure.

Round 3

Reviewer 1 Report

The authors have sufficiently addressed the major concerns. The manuscript can be published.